# Temporal network analysis of gut microbiota unveils aging trajectories associated with colon cancer

Ziqi Chen,[1,2] Zhipeng Zhang,[1] Bei Ning Nie,[3] Wei Huang,[1] Ying Zhu,[2] Long Zhang,[1] Meng Xu,[1] Mengfei Wang,[1] Chenyue Yuan,[1] Ningning Liu,[4] Xinyi Wang,[5] Jianhui Tian,[1] Qian Ba,[2] Ziliang Wang[1,2]

**ABSTRACT** The human gut microbiome's role in colorectal cancer (CRC) pathogenesis has gained increasing recognition. This study aimed to delineate the microbiome characteristics that distinguish CRC patients from healthy individuals, while also evaluating the influence of aging, through a comprehensive metagenomic approach. The study analyzed a cohort of 80 CRC patients and 80 matched healthy controls, dividing participants into a normal and a CRC group, further categorized by age into young, middle-aged, and old-aged subgroups. Extensive metagenomic sequencing of fecal samples allowed for the exploration of both the structural and functional profiles of the microbiome, with findings validated in an independent cohort to ensure robustness. Our results highlight notable differences in microbiome composition between CRC patients and healthy individuals, which exhibit age-dependent variations. Specifically, a higher prevalence of pathogenic bacteria, such as *Bacteroides vulgatus*, known to drive inflammation and carcinogenesis, was observed in CRC patients, alongside a reduction in beneficial microbes, including *Lactobacillus*. Functionally, the CRC-associated microbiome showed an increase in pathways related to DNA repair, cell cycle regulation, and metabolic activities, such as the Citrate cycle and Galactose metabolism, underscoring distinct microbial alterations in CRC patients that could influence disease onset and progression. These insights lay a foundation for future research into microbiome-based diagnostics and treatments for CRC.

**IMPORTANCE** This study underscores the critical role of the gut microbiome in colorectal cancer (CRC) pathogenesis, particularly in the context of aging. By identifying age-specific microbial biomarkers and functional pathways associated with CRC, our findings provide novel insights into how microbiome composition and metabolic activities influence disease progression. These discoveries pave the way for developing personalized microbiome-based diagnostic tools and therapeutic strategies, potentially improving CRC prevention and treatment outcomes across different age groups. Understanding these microbial dynamics could also inform interventions targeting gut microbiota to mitigate CRC risk and progression.

**KEYWORDS** colorectal cancer (CRC), gut microbiome, aging, microbial metabolite

Colorectal cancer (CRC) ranks as the third most prevalent malignant tumor globally and stands as the second leading cause of cancer-related mortality (1). Despite extensive research, a comprehensive understanding of the exact etiology and pathogenesis of CRC remains elusive. Technological advancements, such as high-throughput sequencing, have enabled the identification of alterations in the type and abundance of gut microbiome in CRC patients (2, 3). The intestinal microbiota consists of trillions of bacteria, viruses, fungi, and other microorganisms (4, 5). *Bacteroides*, *Firmicutes*,

Address correspondence to Ziliang Wang, huf_zlwang@126.com, Qian Ba, qba@shsmu.edu.cn, Jianhui Tian, tjhhawk@163.com, or Ziqi Chen, ziqichentcm@shutcm.edu.cn.

Ziqi Chen and Zhipeng Zhang contributed equally to this article. Author order was determined based on their contribution to the article.

The authors declare no conflict of interest.

See the funding table on p. 17.

Portions of this work named "Distinctive microbiome signatures in colorectal cancer: a comparative metagenomic analysis" were presented at the AACR Annual Meeting in 2024, with published abstract number 2807 in Cancer Research.

*Actinobacteria*, *Proteobacteria*, and *Verrucomicrobia* are the predominant bacterial phyla inhabiting the healthy adult colon (6, 7).

Aging further complicates the relationship between the gut microbiome and CRC. CRC incidence rises significantly with advancing age (8), while age-related physiological changes can alter microbial diversity and function (9). Studies have reported that younger CRC patients often exhibit different microbial profiles than older counterparts, suggesting that both host age and microbiome composition may interact to shape CRC risk and progression (10). Characterizing these age-specific microbial shifts is essential for understanding CRC pathogenesis and could inform more personalized treatment approaches.

CRC is a prevalent gastrointestinal malignancy, with its incidence and mortality rates significantly escalating with advancing age (11). Some studies suggest that age is one of the important factors influencing the composition of colonic microbiota in CRC (12). Here are some key findings: decreased microbial diversity: as individuals age, the diversity of intestinal microbiota tends to decrease in CRC patients (13). This decline may be related to aging of intestinal function and changes in the immune system (14). Changes in abundance of specific bacterial groups: there is an association between age and relative abundance of certain bacterial groups. For example, some studies have found that increasing age is associated with a decrease in beneficial bacteria (such as *Clostridium*) and an increase in pathogenic bacteria (such as *Enterococcus*) (15). Changes in inflammatory status: age-related changes in intestinal microbiota may lead to alterations in inflammatory status (16). Inflammation plays a crucial role in the occurrence and development of ; CRC therefore, age-related changes in microbial communities may indirectly affect the risk of colon cancer (17). As individuals age, they experience a range of physiological and metabolic changes that have the potential to impact both the composition and functionality of their gut microbiota (18). Research has identified notable disparities in the gut microbiota composition among CRC patients belonging to distinct age cohorts (19).

Younger patients typically exhibit greater diversity within their gut microbiota. For instance, bacteria from the *Bacteroides* genus are prevalent among younger individuals but notably diminished within older patient populations. These alterations could potentially stem from age-associated physiological and metabolic attributes (20).

In a comparative study, the gut microbiota of elderly CRC patients was found to exhibit a significant reduction in diversity compared to healthy elderly individuals. Furthermore, specific alterations were observed in bacterial populations, including a notable decrease in beneficial gas-producing *Bifidobacterium* species derived from plant fiber digestion and an increase in potentially harmful bacteria associated with tumors or pathogens. These findings suggest that age-related changes in the composition and function of gut microbiota in CRC patients are associated with the initiation and progression of the disease (21).

In this study, we used a comprehensive metagenomic approach to investigate gut microbiome alterations in CRC patients versus healthy controls, stratified by age groups. We hypothesized that CRC-associated dysbiosis would exhibit distinct patterns across different ages, potentially informing new avenues for microbiome-based strategies in CRC prevention and therapy. By comparing multiple cohorts and validating our findings externally, we provide insights that could guide more tailored interventions—ultimately aiming to reduce CRC incidence and improve outcomes across all age groups.

Overall, the following exploration underscores the critical interplay between age, gut microbiota composition, and CRC pathogenesis. A deeper understanding of these relationships may facilitate personalized interventions that leverage microbiome-targeted diagnostics and therapeutics to improve patient outcomes.

## MATERIALS AND METHODS

### Data profile

We categorized data into two main groups: normal and colon cancer. Based on age, we further divided these into three subgroups: <50 years, ≥50 to <65 years, and ≤65 years, which were labeled as the young group, middle-aged group, and old-aged group, respectively. Among them, our own data set PRJNA731589 consisted of 12 healthy individuals in the young group. Additionally, there were 16 colon cancer patients in the young CRC group, 52 healthy individuals in the normal middle-aged group, and 36 colon cancer patients in the middle-aged CRC group. Furthermore, there were 20 healthy individuals in the normal old age group along with 27 old age colon cancer patients. For validation purposes, we included a separate data set, PRJNA763023, consisting of a total of 100 individuals from the young age group; among them, half had colon cancer while half were healthy. Similarly, for both middle-aged and old age groups combined together, we used data set PRJNA763023, comprising a total of 200 individuals; out of these, 100 had colon cancer while the remaining 100 were healthy (Table 1).

### The analysis of metagenome sequencing data

The main execution software employed in this study included fastqc (v0.12), trimmomatic (v0.39), and bowtie2 (v2.3.5.1). These tools were utilized for quality control, filtering, and comparison of the data sets separately.

- Trimmomatic was used to remove low-quality reads (SLIDINGWINDOW: 4:20 MINLEN: 50).
- Bowtie2 v2.3.5 was applied to eliminate matched read lengths that potentially originated from host-associated or lab-associated sequences, using the human reference database.
- MetaPhlAn v4.0.6 was utilized to calculate relative abundance based on the mpa_vOct22_CHOCOPhlAnSGB_202212 database.
- Functional compositional analysis was performed using human3 v3.8 with the full_chocophlan.v296_201901b database.
- To minimize potential false positives, taxonomic profiles with relative abundance values <0.1% in the samples were filtered out.
- For downstream analyses, species abundance data from each data set were integrated with relevant metadata using the Phyloseq package in R.
- Alpha and beta diversity indices for gut microbiome analysis were estimated based on species' relative abundance profiles using the Vegan package in R and visualized using ggplot2.

### The identification of age-specific microbial communities of CRC or healthy people

In this study, a combination of statistical, machine learning, and network-based methods was employed to identify potential biomarkers for CRC. Specifically, the multivariate associative linear modeling algorithm (MaAsLin2) was utilized to control for confounding factors such as gender and body mass index (BMI), while data standardization was performed using the trimmed mean of M values (TMM) assay. Six analysis methods including *t*-test, Wilcoxon test, linear discriminant analysis of effect size (LEfSe), Deseq2, Edger, and MaAsLin2 were selected for variance analysis. Four microbial communities showing differential abundance between CRC patients and healthy individuals

TABLE 1  Cohorts of data set PRJNA731589 and validation set PRJNA763023[a]

| Data set | CRC young | Normal young | CRC middle-aged | CRC old-aged | Normal middle-aged | Normal old-aged |
|---|---|---|---|---|---|---|
| PRJNA731589 | 16 | 12 | 36 | 27 | 52 | 20 |
| PRJNA763023 | 50 | 50 | 100 | | 100 | |

[a]Cohort: young: <50 years, middle-aged: ≥50 to <65 years, and old-aged: ≤65 years.

at different ages were identified as clusters based on FDR-adjusted $P \leq 0.05$ criteria. Additionally, LEfSe considered microbial communities with effect size LDA scores >4 and FDR-adjusted $P \leq 0.05$ as significantly different. Further analyses focused on the family level and genus level.

Three machine learning algorithms, namely, random forest (RF), support vector machine (SVM), and generalized linear model (GLM), were applied in this study. RF was chosen for its ability to handle high-dimensional data and its robustness to overfitting, making it particularly suitable for microbiome data analysis. SVM was selected for its effectiveness in classification tasks with complex data sets, while GLM was used to model the relationship between microbial abundance and clinical outcomes. Among these, RF demonstrated superior classification performance, as evidenced by its higher accuracy in cross-validation trials. RF demonstrating superior classification performance. To identify the most informative microbial biomarkers, five 10-fold cross-validation trials were conducted using RF, followed by the selection of critical points based on averaging minimum cross-validation errors. The most discriminatory biomarkers were determined by averaging decreasing accuracy scores obtained from RF modeling characteristic importance scores in relation to phenotype.

To explore microbial correlations further within a co-abundance network framework utilizing the ggCLusterNet package, SparCC was employed to estimate the microbial correlation matrix, which facilitated the construction of modular microbial co-abundance networks. Correlations with FDR-adjusted $P$ value < 0.05 and magnitude > 0.6 were selected for subsequent visualization and network analysis. The layout was visualized by the igraph package.

## Species and function associations

The Spearman correlation coefficients were separately calculated for the CRC/healthy group and adjusted for FDR multiple testing using the Benjamini and Hochberg (BH) method. Only correlations with absolute values of correlation coefficients > 0.5 and $P$ values < 0.05 were retained. Correlation heatmaps were generated using the corrplot software package.

## RESULTS

### A pronounced disparity was found in distributions among CRC and healthy groups within individual age categories

A comparative analysis of fecal microbial alpha diversity measures was performed to evaluate bacterial richness across cohorts. Our analysis revealed that the richness indices were substantially lower in both the normal middle-aged group and CRC middle-aged group than in the Normal young group, indicated by a notable reduction in the number of observed operational taxonomic units. On the other hand, the fecal microbial Simpson's evenness index indicated a significant increase in the CRC young group and CRC old groups compared to the normal young group, suggesting improved evenness within the gut microbiome of CRC cohorts (Fig. 1A). Notably, Pielou's evenness index and Shannon diversity index did not exhibit substantial variation among the groups. Using principal coordinate analysis (PCoA) based on Bray-Curtis dissimilarity and verified by permutation multivariate analysis of variance (PERMANOVA), we found a significant difference in microbiota structure distribution between the CRC and healthy cohorts (Fig. 1B). Moreover, there was a pronounced disparity in distributions among CRC and healthy groups within individual age categories (Fig. 1C and D). These disparities suggest that each group harbors a distinct gut microbial diversity and structure. The bacterial Venn diagrams and UpSet plots further support this, showcasing a greater number of unique microbial species in the healthy group compared to the CRC group, with the normal middle-aged group exhibiting the fewest (Fig. 1E and F). A robust machine learning framework was utilized to evaluate the predictive power of the classification models at various taxonomic levels, with results displayed in Fig. S1. The analysis revealed an

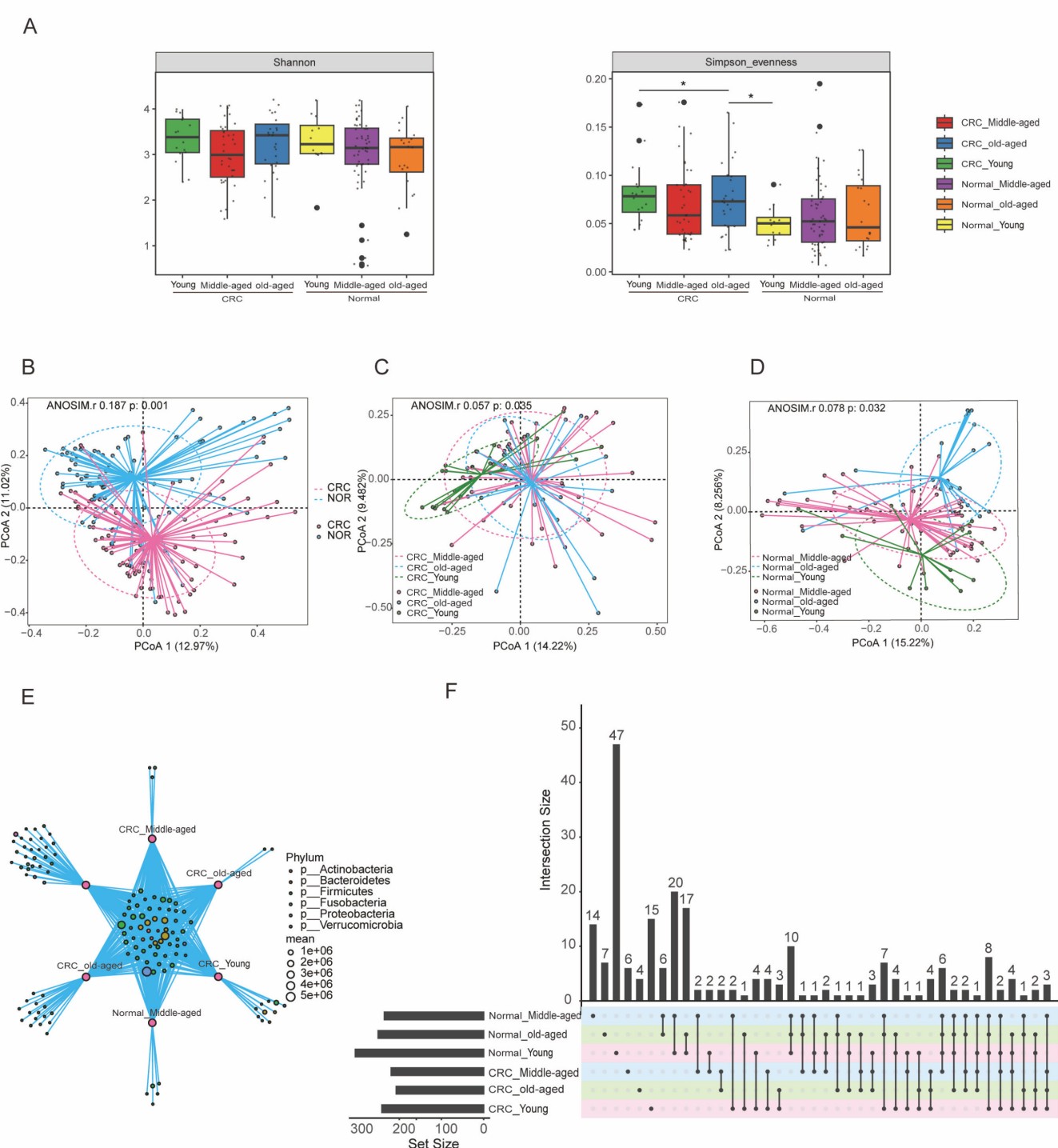

**FIG 1** (A) α-Diversity estimated by the Pieiou evenness, Shannon, Simpson evenness index, *P* value calculated by two-sided unpaired *t*-test. (B) Principal coordinate analysis (PCoA) of the diseased and healthy groups based on the Bray-Curtis distance, which showed that microbial composition was significant between groups (*P* = 0.001) and between cohorts (*P* = 0.001). *P* values for β-diversity based on Bray-Curtis distance were calculated using PERMANOVA by two-sided test. (C) PCoA based on Bray-Curtis distances for age-specific CRC groups. (D) PCoA based on Bray-Curtis distance for healthy groups at different ages. (E) Wayne plot showing overlap between groups. (F) Upset plots showing the number of differential bacterial species in each group, shared by combinations of data sets. The numbers above each column indicate the size of the differential species. The set size on the right indicates the number of differential species in each cohort.

upsurge of taxonomic groups such as *Bacteroidia*, *Bacteroidetes*, *Bacteroidales*, *Bacteroidaceae*, *Bacteroides*, *Phocaeicola*, and *Phocaeicola vulgatus* in the CRC groups relative to healthy counterparts across all ages. Conversely, *Gammaproteobacteria*, *Firmicutes*, *Proteobacteria*, *Enterobacterales*, *Enterobacteriaceae*, *Bacteroides*, *Prevotella copri clade A*, and *Klebsiella pneumoniae* were noted to decline in the CRC groups compared to their healthy age-matched groups.

## Phylogenetic characterization of fecal microbial communities in colon cancer within individual age categories

Next, we conducted phylogenetic characterization of fecal microbial communities associated with colon cancer, selecting only those resembling more than 10% of the groups. We utilized a MaAsLin2 to controvert the gender and BMI factors, followed by data standardization using the TMM method. Six types of tests were administered: *t*-test, Wilcoxon test, LEfSe, Deseq2, Edger, and MaAsLin2. These were employed for the variance analyses. The groups selected, based on frequent appearances in the test findings, indicated differentiation between healthy individuals and CRC patients in diverse age groups (Fig. S2). Successive differential analyses identified 31 bacterial genera and 45 species variation among the CRC samples collected from three age groups. The younger age group displayed biomarkers such as *Parasutterella excrementihominis*, *Anaerotruncus colihominis*, and *Bacteroides cellulosilyticus*; the middle age group showed *Coprococcus comes*, *Fusicatenibacter saccharivorans*, and *Citrobacter freundii*; and the older age group suggested *Blautia obeum*, *Roseburia intestinalis*, and *Bacteroides eggerthii* (Fig. 2A through F). Examination of genus-level and species-level gut microbiota with identical trends and differentiation levels conducted on the PRJNA763023 data set (Table 2) revealed an increase in *Bacteroides*, *Bilophila*, and *Fusobacterium* in the CRC groups across all age clusters, both at the genus and species gradation (Table 3), where *Bacteroides thetaiotaomicron*, *Bilophila wadsworthia*, and *Clostridium symbiosum* showed increase.

## Phylogenetic characterization of fecal microbial communities in different age groups with colon cancer

In this study, phylogenetic characterization was performed on fecal microbial communities from three age cohorts with colon cancer, focusing on 31 genera and 45 species. Analysis revealed six bacteria with notable discriminatory potential associated with aging within the CRC group (Fig. 3A and C). The genera *Coprococcus* and *Roseburia* demonstrated a decrease in abundance with advancing age, whereas *Fusicatenibacter*, *Parabacteroides*, and the species *C. comes*, *F. saccharivorans*, and *P. excrementihominis* increased in prevalence with age. In contrast, seven bacterial indicators showed substantial age-related changes in the healthy control group (Fig. 4A and C), with *Coprococcus* declining and *Erysipelatoclostridium*, *Fusicatenibacter*, *Lachnoclostridium*, *Bacteroides nordii*, *B. obeum*, and *R. intestinalis* all increasing in abundance with age. Comparable patterns were observed in the data set PRJNA763023 (Fig. 3B and 4B).

## The construction of co-abundance networks

In order to understand the potential interactions among various age groups of gut microbial communities and their possible role in the pathogenesis of CRC, we carried out a co-abundance association analysis. Among the groups studied, the young CRC group demonstrated 14,048 associations across 695 species (Fig. 5A), the middle-aged CRC group had 6,633 associations for 714 species (Fig. 5B), and the old CRC group reported 10,791 associations among 752 species (Fig. 5C). Additionally, the young healthy group displayed 20,715 associations for 872 species (Fig. 5D), 5,059 associations were found for 622 species in the middle-aged healthy group (Fig. 5E), and 10,091 associations were observed for 678 species in the old healthy group (Fig. 5F). Overall, the young age group's networks appeared to be more complex and interconnected, while those of the middle-aged group were less connected. A decrease in correlations was observed in the

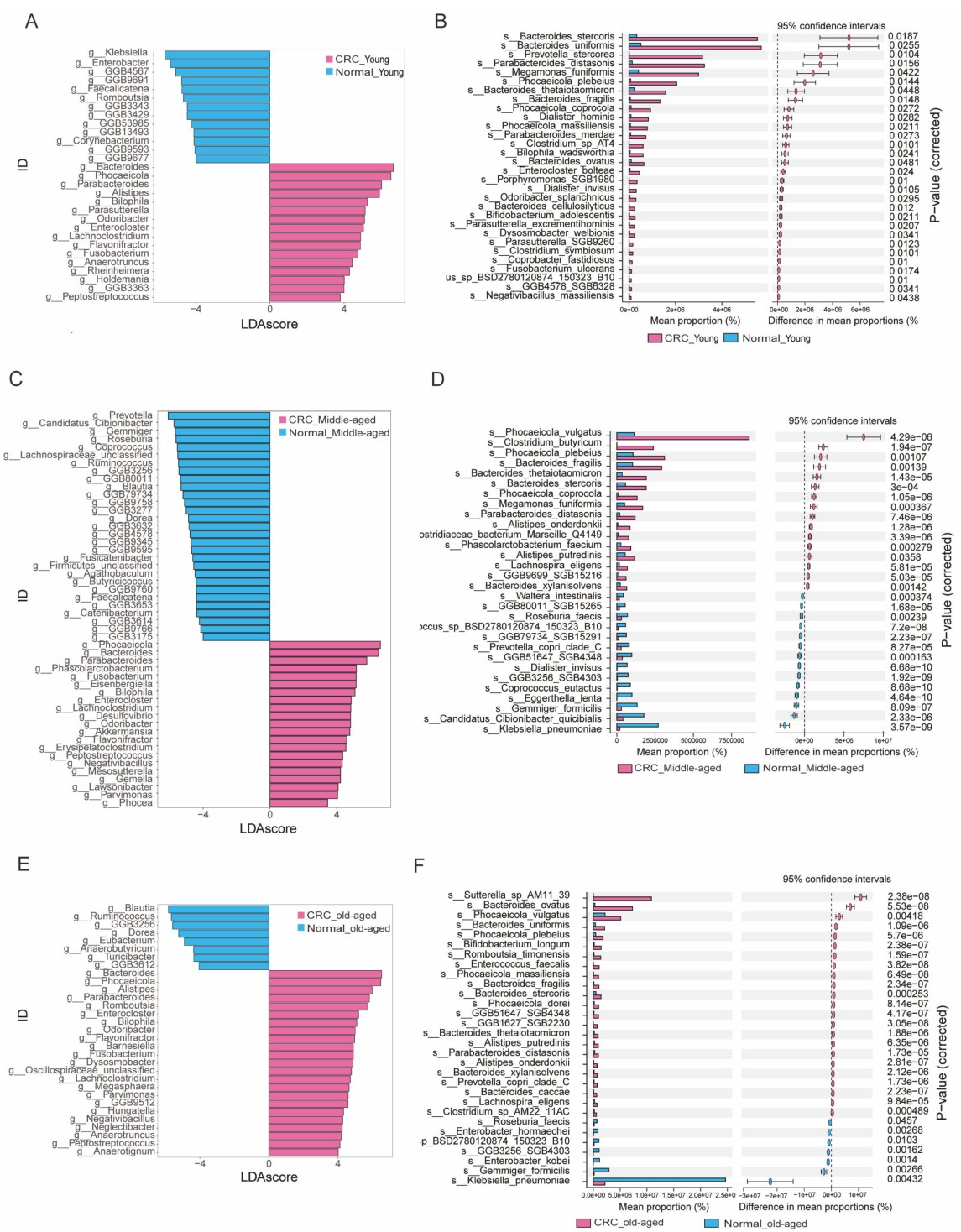

**FIG 2** (A) Histogram of LDA at species level for the CRC young group versus the healthy young group (LDA score > 3.0, *P* < 0.05). (B) Stamp plots at genus level for the CRC young group versus the healthy young group (two-sided unpaired *t*-test, *P* < 0.05). (C) Histogram of LDA at the genus level of the CRC middle-aged group versus the healthy middle-aged group (LDA score > 3.0, *P* < 0.05). (D) Stamp plots of the genus level of the CRC middle-aged group versus the healthy middle-aged group (two-sided unpaired *t*-test, *P* < 0.05). (E) Histogram of LDA at the genus level of the elderly CRC group vs the healthy old-aged group (LDA score > 3.0, *P* < 0.05). (F) Stamp plots at the genus level of the geriatric CRC group versus the geriatric healthy group (two-sided unpaired *t*-test, *P* < 0.05).

young CRC network when compared to the healthy group, but an increase was noted in the CRC network of both the middle-aged and old-aged groups.

TABLE 2 Genus-level gut microbiota with the same trend and differential levels in own data and validation set PRJNA763023[a]

| Gut microbiota | Young CRC versus normal | | Middle-aged CRC versus normal | | Old-aged CRC versus normal | |
|---|---|---|---|---|---|---|
| | Up | Down | Up | Down | Up | Down |
| g__Agathobaculum | | | | * | | |
| g__Alistipes | * | | | | | |
| g__Anaerobutyricum | | | | | | * |
| g__Anaerotruncus | * | | | | | |
| g__Bacteroides | * | | * | | * | |
| g__Bilophila | * | | * | | * | |
| g__Blautia | | | | * | | * |
| g__Butyricicoccus | | | | * | | |
| g__Candidatus_Nanosynsacchari | | | | | | * |
| g__Coprobacter | * | | | | | |
| g__Coprococcus | | | | * | | |
| g__Dorea | | | | | | * |
| g__Erysipelatoclostridium | | | * | | | |
| g__Eubacterium | | | | * | | |
| g__Faecalicatena | | * | | | | |
| g__Firmicutes_unclassified | | | | * | | |
| g__Flavonifractor | * | | | | * | |
| g__Fusicatenibacter | | | | * | | * |
| g__Fusobacterium | * | | * | | * | |
| g__Gemella | | | * | | | |
| g__Lachnoclostridium | * | | | | * | |
| g__Megasphaera | | | | | * | |
| g__Parabacteroides | | | * | | * | |
| g__Parasutterella | * | | | | | |
| g__Parvimonas | * | | | | * | |
| g__Peptostreptococcus | | | * | | * | |
| g__Raoultella | | | | | | * |
| g__Romboutsia | | | | | | * |
| g__Roseburia | | | | * | | |
| g__Ruminococcus | | | | | | * |
| g__Turicibacter | | | | | | * |

[a]Data and validation set PRJNA763023; grouping: CRC young versus normal young; CRC middle-aged versus normal middle-aged; CRC old-aged versus normal old-aged.

## Functional analysis of the fecal microbiota

In the subsequent functional analysis of fecal microbiota, we assessed alterations in the Kyoto Encyclopedia of Genes and Genomes (KEGG) gene and pathway levels, identifying 17 distinct modules (Fig. 6A). Our metagenomic differential analysis of the KEGG database elucidated a marked enrichment of metabolic pathways within the CRC groups. Notably, pathways including glyoxylate and dicarboxylate metabolism, biosynthesis of cofactors, and lipopolysaccharide biosynthesis were highly represented, as denoted by extended bars and redder hues in the visualization, implicating their significance in energy generation, molecular building block synthesis, and cellular structure establishment (Fig. 6B). Additionally, the "Carbon metabolism" and "Glycine, serine, and threonine metabolism" pathways evidenced significant gene enrichment, indicative of intense carbon utilization and amino acid biosynthesis operations within the microbial communities analyzed.

**TABLE 3** Species-level gut microbiota with the same trend and difference levels in own data and validation set PRJNA763023[a]

| Gut microbiota | Young CRC versus normal | | Middle-aged CRC versus normal | | Old-aged CRC versus normal | |
|---|---|---|---|---|---|---|
| | Up | Down | Up | Down | Up | Down |
| s__Agathobaculum_butyriciproducens | | | | * | | |
| s__Anaerobutyricum_hallii | | | | | | * |
| s__Anaerotruncus_colihominis | * | | | | | |
| s__Bacteroides_cellulosilyticus | * | | | | | |
| s__Bacteroides_eggerthii | | | | | * | |
| s__Bacteroides_fragilis | | | * | | * | |
| s__Bacteroides_nordii | | | * | | * | |
| s__Bacteroides_ovatus | * | | | | * | |
| s__Bacteroides_thetaiotaomicron | * | | * | | * | |
| s__Bifidobacterium_bifidum | * | | | | | |
| s__Bilophila_wadsworthia | * | | * | | * | |
| s__Blautia_faecis | | | | * | | * |
| s__Blautia_massiliensis | | | | | | * |
| s__Blautia_obeum | | | | * | | * |
| s__Blautia_stercoris | | | | | | * |
| s__Blautia_wexlerae | | | | * | | * |
| s__Citrobacter_freundii | | | | * | | |
| s__Clostridiales_bacterium | * | | | | * | |
| s__Clostridium_butyricum | | | * | | | |
| s__Clostridium_symbiosum | * | | * | | * | |
| s__Coprococcus_comes | | | | * | | |
| s__Coprococcus_eutactus | | | | * | | |
| s__Dialister_pneumosintes | | | * | | | |
| s__Dorea_formicigenerans | | | | * | | * |
| s__Dorea_longicatena | | | | | | * |
| s__Dysosmobacter_welbionis | * | | | | | |
| s__Enterobacter_hormaechei | | | | * | | |
| s__Enterocloster_bolteae | * | | * | | | |
| s__Erysipelatoclostridium_ramosum | | | * | | | |
| s__Faecalicatena_fissicatena | | | | | | * |
| s__Flavonifractor_plautii | * | | | | * | |
| s__Fusicatenibacter_saccharivorans | | | | * | | * |
| s__Lachnospiraceae_bacterium | | | | * | | |
| s__Parabacteroides_distasonis | | | * | | * | |
| s__Parabacteroides_faecis | | | | | * | |
| s__Parabacteroides_merdae | * | | | | | |
| s__Parasutterella_excrementihominis | * | | | | | |
| s__Parvimonas_micra | | | | | * | |
| s__Peptostreptococcus_stomatis | | | * | | | |
| s__Prevotella_intermedia | | | | | * | |
| s__Romboutsia_timonensis | | | | | | * |
| s__Roseburia_hominis | | | | * | | |
| s__Roseburia_intestinalis | | | | | | * |
| s__Ruminococcus_callidus | | | | * | | * |
| s__Streptococcus_salivarius | | * | | | | |

[a]Data and validation set PRJNA763023; grouping: CRC young versus normal young; crc middle-aged versus normal middle-aged; crc old-aged versus normal old-aged.

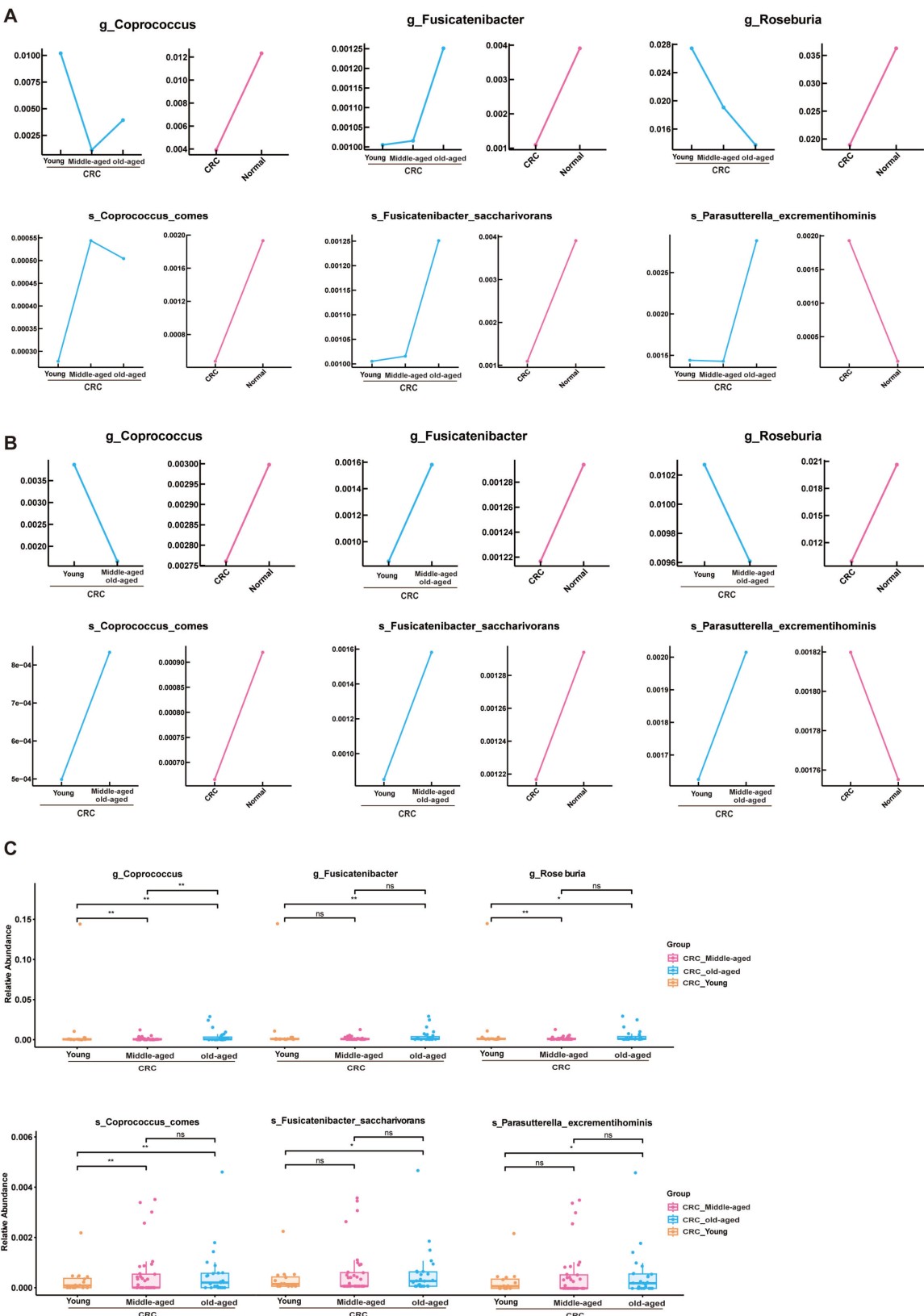

**FIG 3** (A) CRC group with significant change in age group in PRJNA731589. (B) Line graph of CRC groups with significant changes in age-specific flora in validation set PRJNA763023. (C) CRC group with significant change in age group in PRJNA731589 box plot showing significant change.

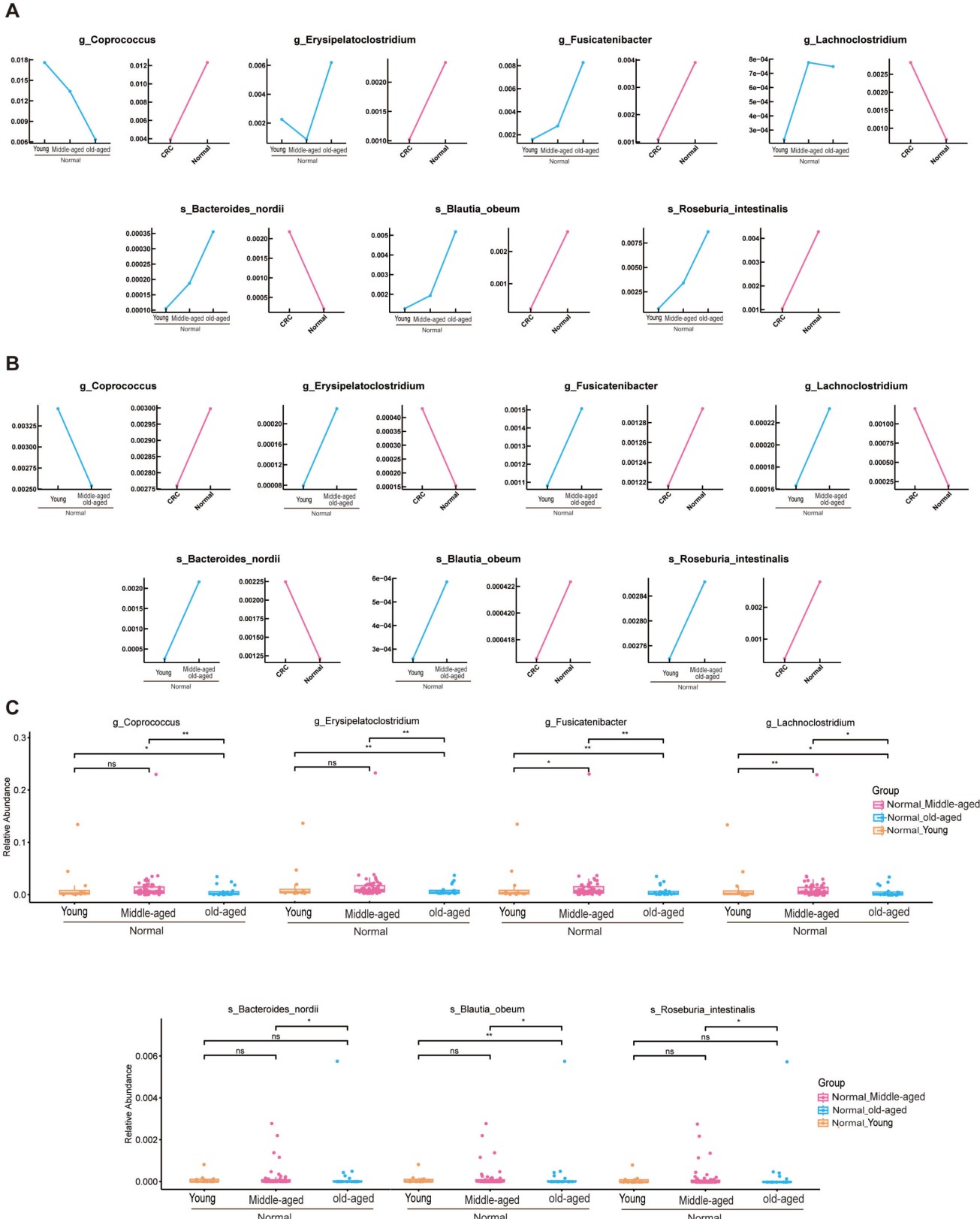

**FIG 4** (A) Line graph of the healthy group with significant changes in age groups in PRJNA731589. (B) Line graph of the healthy group with significant changes in age-specific flora in the validation set PRJNA763023. (C) Healthy group at different ages with significant change in flora in PRJNA731589. Box line plot showing significant change.

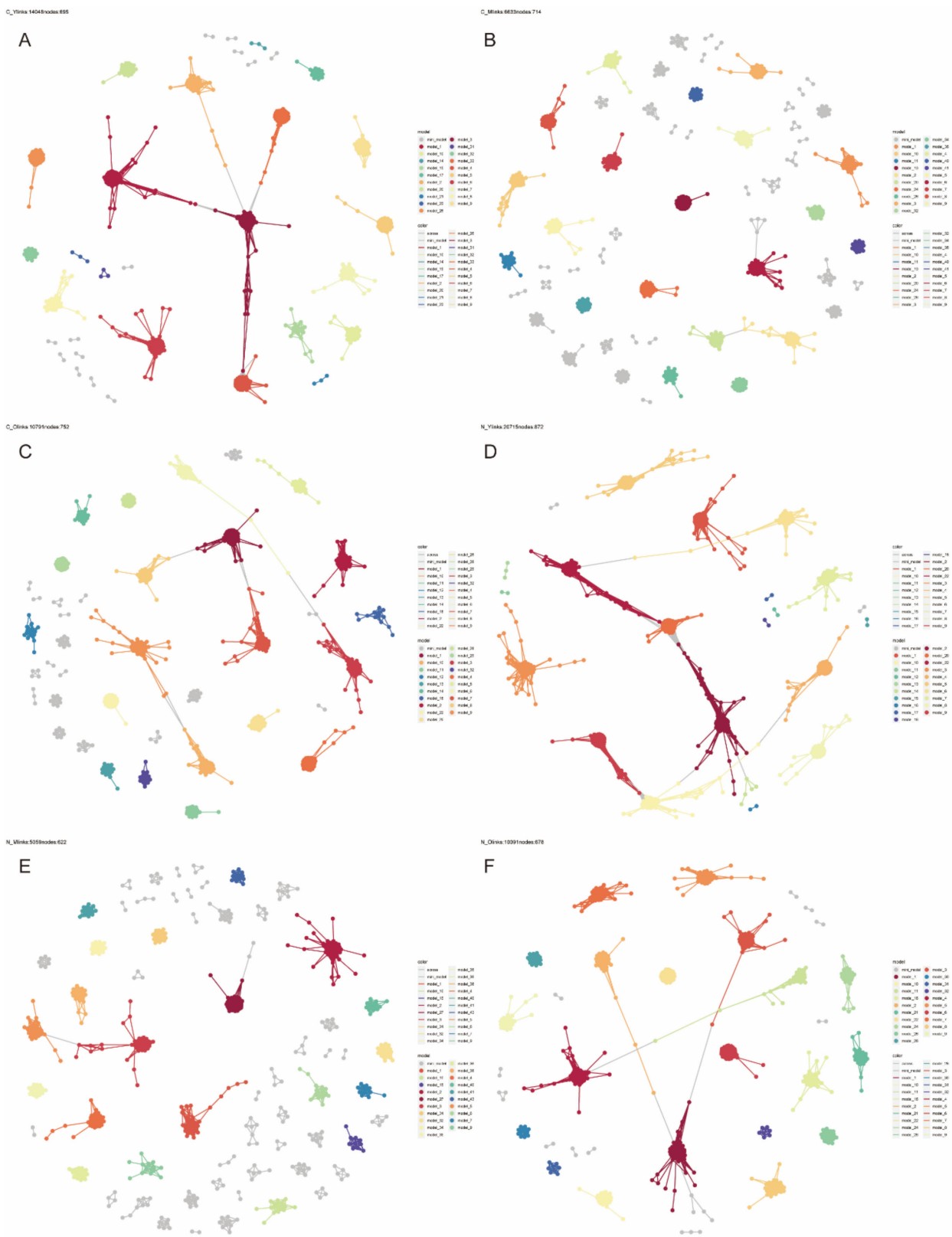

**FIG 5** (A) Co-abundance network in the young disease group. (B) Co-abundance network for the middle-aged CRC group. (C) Co-abundance network for the older CRC group. (D) Co-abundance network for the young healthy group. (E) Co-abundance network for the middle-aged healthy group. (F) Co-abundance network in the elderly healthy group. Absolute correlation higher than 0.8, FDR < 0.05 significance threshold.

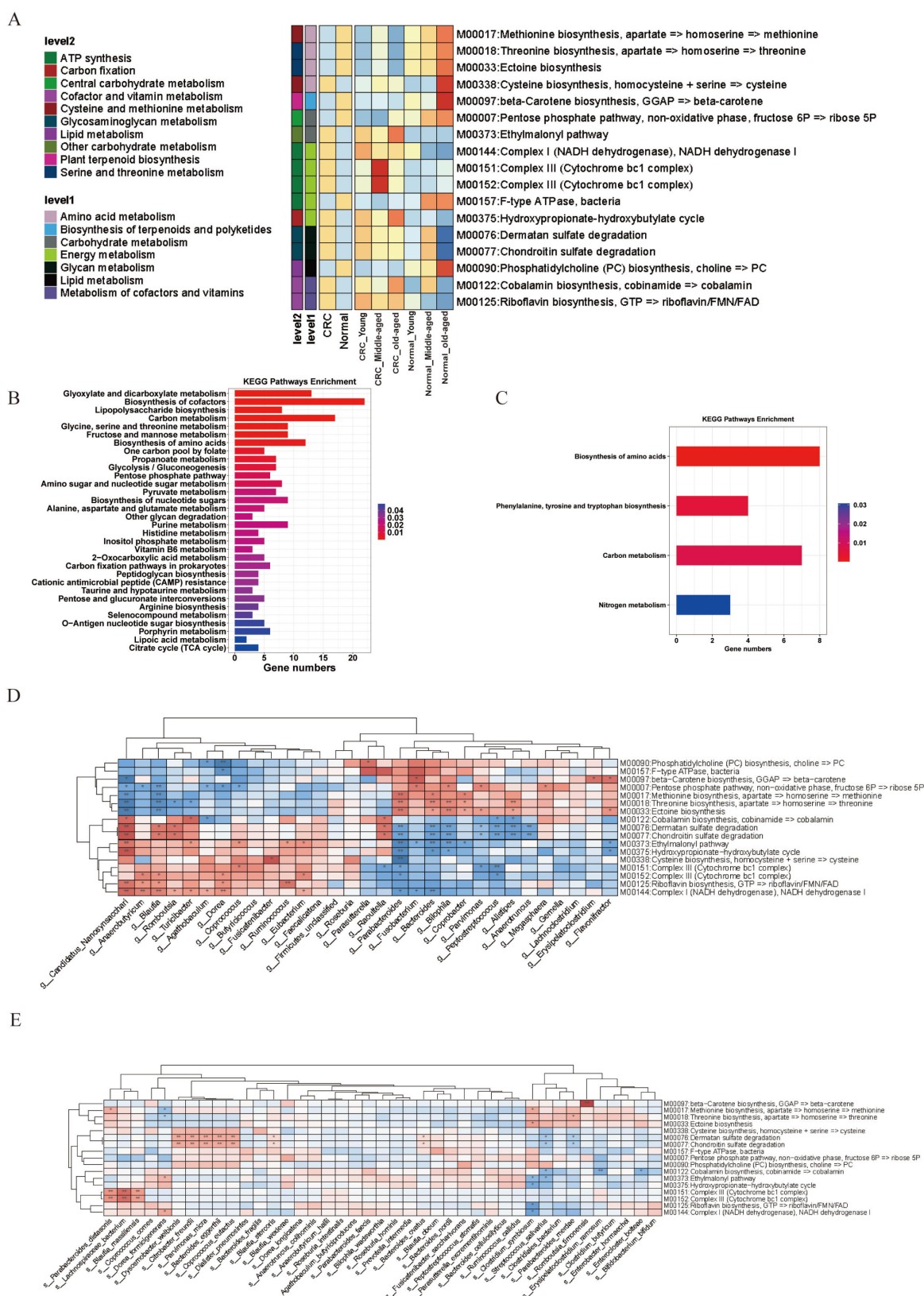

FIG 6   (A) Differential model heatmap in healthy versus CRC groups. (B) KEGG age-differentiated pathways enriched in the CRC group. (C) KEGG age-differentiated pathways enriched in the healthy group. (D) Heatmap of species-level hub colonization with differential module correlations. (E) Heatmap of genus-level hub colonies with differential module correlations. The bar chart provided illustrates the number of genes associated with each pathway, with the length of the bar indicating the gene count and the color gradient representing the significance of enrichment, from less significant (blue) to more significant (red).

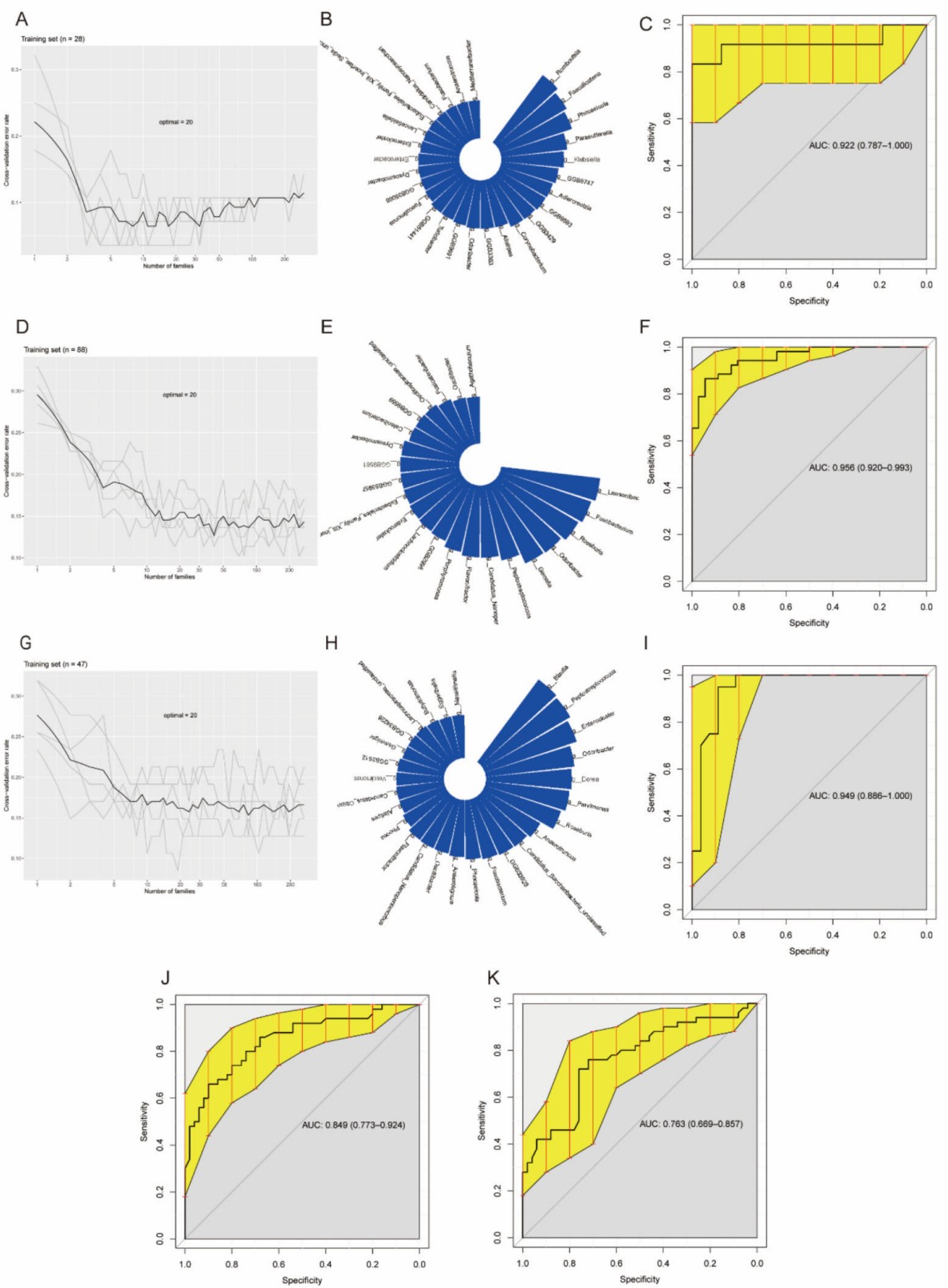

**FIG 7** (A) Tenfold cross-validation of the random forest model between the young CRC and young healthy groups during the training phase. (B) Circle plots of the top 30 differentially abundant markers selected as the best set of markers for the young CRC group and the young healthy group based on the random forest. (C) AUC values of the random forest model for the young CRC group and the young healthy group, and the error lines indicate the 95% confidence

**Fig 7 (Continued)**

intervals of the AUC values. (D) Tenfold cross-validation of the random forest model between the middle-aged CRC group and the middle-aged healthy group during the training phase. (E) Circle plots of the top 30 differential abundance markers selected as the best set of markers for the middle-aged CRC and middle-aged health groups based on the random forest. (F) AUC values of the random forest model for the middle-aged CRC group and middle-aged health group, and the error lines indicate the 95% confidence intervals of the AUC values. (G) Random forests between the old age CRC group and the old age health group during the training phase. The model was cross-validated 10-fold. (H) Circle plots of the top 30 differentially abundant markers selected as the best set of markers for the geriatric CRC group and geriatric health group based on random forest. (I) AUC values of the randomized forest model for the elderly CRC group and the elderly health group; error lines indicate 95% confidence intervals of the AUC values. (J) AUC values based on the random forest model for the young CRC group and the young health group for the young sample (<50 years old) in the external independent validation set PRJNA763023. (K) AUC values based on the random forest model for the middle-aged and elderly samples (>50 years old) in the external independent validation set PRJNA763023, based on the middle-aged and elderly CRC group and middle-aged and elderly health group.

The observed upregulation of the "Fructose and mannose metabolism" and "Biosynthesis of amino acids" pathways points to vigorous carbohydrate metabolism and protein production activities. Furthermore, gene enrichment in the "Citrate cycle (TCA cycle)" underscores a pivotal role in energy derivation via aerobic respiration. Genes associated with nucleotide sugar metabolism, specifically the "Amino sugar and nucleotide sugar metabolism" and "Biosynthesis of nucleotide sugars" pathways, also emerged prominently, underscoring their criticality in the synthesis of structural polysaccharides and glycoconjugates. Conversely, the healthy group demonstrated fewer differentially enriched metabolic pathways, with a focus on amino acid biosynthesis, carbon metabolism, phenylalanine, tyrosine, and tryptophan biosynthesis, and nitrogen metabolism (Fig. 6C).

We identified 17 differential modules, notably the Phosphatidylcholine (PC) biosynthesis, significantly attributed to the age-related differential bacterial genera such as *Parasutterella*, *Agathobaculum*, and *Dorea*. Genera including *Parabacteroides*, *Bacteroides*, and *Bilophila* revealed substantial biosynthetic contributions toward Threonine and Ectoine. The genus *Candidatus Nanosynsacchari* played a leading role in several biosynthetic pathways, including Cobalamin biosynthesis and Dermatan sulfate degradation, while *Blautia* was markedly involved in the Pentose phosphate pathway and Methionine biosynthesis. Additionally, species such as *Parabacteroides distasonis* and *Lachnospiraceae bacterium* contributed to the respiratory Complex III, with *Dysosmobacter welbionis* and *Citrobacter freundii* showing significant enrichment in Dermatan sulfate and Chondroitin sulfate degradation (Fig. 6D and E), reflecting the diverse functionality and importance of distinct microbial communities. Genus-level and species-level differences and corresponding hierarchical bar graphs are depicted in Fig. S3 and S4.

In conclusion, through metagenomic differential analysis, we have spotlighted central metabolic pathways abundantly expressed in CRC-associated microbial communities, elucidating their functional proficiencies and likely bacterial adaptations within the gut ecosystem.

## Identification and validation of markers based on fecal microbial community

Analysis was conducted specifically on the family and genus levels using three machine learning methods: RF, SVM, and GLM. RF demonstrated the most powerful classification ability (refer to Fig. 7). Five iterations of 10-fold cross-validation trials were performed using RF to ascertain optimal microbial biomarkers and pinpoint crucial junctures by averaging minimum cross-validation errors. Biomarkers exhibiting the most significant discrimination were selected using an average decreasing accuracy (feature importance scores in RF) and were considered to yield the least error. The outcome showed the aged model had significant accuracy levels of AUC: 0.922 for young, AUC: 0.956 for middle-aged, and AUC: 0.949 for old groups. Moreover, the correlation model classified participants into two age groups from the external validation set at 50 years, with AUC

levels reporting 0.849 for the younger participants and 0.763 for the middle-aged and elderly participants (refer to Fig. 7).

## DISCUSSION

In this study, we found that the intestinal microbiota of patients with CRC is closely related to intestinal metabolism and aging.

The abundance of specific microbiota shows a certain positive or negative correlation with intestinal metabolic activity and aging. This suggests that the intestinal microbiota affects the development of CRC by regulating metabolic pathways and the process of intestinal aging. This discovery deepens our understanding of the role of the intestinal microbiota in the development of CRC.

Our results suggest that the intestinal microbiota in CRC patients exhibits pathogenic properties. Specific microbial communities likely promote tumor cell growth and energy metabolism, potentially accelerating the intestinal aging process. This provides important evidence for the role of the intestinal microbiota in the treatment and prevention of CRC (22).

Furthermore, our research has identified some metabolic pathways and modules associated with the intestinal microbiota by regulating aging. These molecules may be key factors in the regulation of intestinal metabolism and aging, and in the future, interventions can be developed to modulate these molecules to prevent the development of CRC (23).

Our findings are consistent with previous studies that have reported a higher prevalence of pathogenic bacteria such as *Bacteroides vulgatus* in CRC patients. For instance, a study conducted in China by Chen et al. (24) also observed an increase in Bacteroides species in CRC patients, particularly in older age groups. However, our study uniquely identified age-specific microbial biomarkers, such as *P. excrementihominis* in younger patients and *B. obeum* in older patients, which were not reported in previous studies. These differences may be attributed to variations in dietary habits and environmental factors across different geographic regions.

Furthermore, we explored the potential association between differential microbial communities and clinical features such as tumor stage and patient prognosis. For instance, the increased abundance of *B. vulgatus* in CRC patients was correlated with advanced tumor stages, suggesting a potential role of this pathogen in tumor progression (25). However, further studies are needed to elucidate the causal relationships between specific microbial taxa and clinical outcomes in CRC patients. Our study identified distinct microbial biomarkers across different age groups of CRC patients. In younger patients, *P. excrementihominis* and *Anaerotruncus colihominis* were significantly enriched, while in older patients, *B. obeum* and *R. intestinalis* were more prevalent. These age-specific microbial changes may reflect the physiological and metabolic alterations associated with aging, such as reduced immune function and changes in dietary habits (26, 27). While our young CRC cohort is small, we have mitigated this issue through rigorous statistical validation and cross-referencing with external data sets. Additionally, the consistency of our findings with results from larger cohorts in the validation data set supports the reliability of our conclusions. Understanding these age-related microbial shifts could provide insights into personalized therapeutic strategies for CRC patients across different age groups.

Our research illuminates the intestinal microbiota's role in regulating tumor metabolism and aging. Future investigations should explore the underlying mechanisms connecting intestinal microbiota with host metabolism and aging, potentially leading to microbiota-based preventive and therapeutic strategies for CRC.

## ACKNOWLEDGMENTS

The work was supported by the Young talents project Talents Project of Shanghai Municipal Health Commission (2022YQ066), the National Natural Science Foundation of China (82304777), the Shanghai Sailing Program of Science and Technology

Innovation Plan of Shanghai Science and Technology Commission (23YF1442800), and the Traditional Chinese Medicine Science and Technology Development Project of Shanghai Medical Innovation & Development Foundation (WL-HBQN-2021004K, WL-QNPY-2022004K,WL-BJRC-2022001K), and the Open Foundation of Shaanxi University of Chinese Medicine Key Laboratory of Research & Development of Characteristic Qin Medicine Resources (KF202330).

Z.C. conceived, analyzed, and wrote the article. Z.Z. and B.N. helped to analyze the data. Y.Z., L.Z., M.X., W.H., M.W., Z.Z., C.Y., N.L., and X.W. provided technical assistance. J.T.: Conceptualization, Supervision. Q.B.: Conceptualization, Writing—Reviewing and Editing. Z.W.: Conceptualization, Writing—Reviewing and Editing, Supervision. All authors read and approved the final manuscript.

## AUTHOR AFFILIATIONS

[1]Institute of Oncology, Shanghai Municipal Hospital of Traditional Chinese Medicine, Shanghai University of Traditional Chinese Medicine, Shanghai, China
[2]Laboratory Center, Shanghai Municipal Hospital of Traditional Chinese Medicine, Shanghai University of Traditional Chinese Medicine, Shanghai, China
[3]Nanjing University of Traditional Chinese Medicine, Nanjing, China
[4]State Key Laboratory of Systems Medicine for Cancer, Center for Single-Cell Omics, School of Public Health, Shanghai Jiao Tong University School of Medicine, Shanghai, China
[5]Department of Pathology, University of California, San Diego, La Jolla, California, USA

## AUTHOR ORCIDs

Ziqi Chen http://orcid.org/0009-0005-5995-6546
Jianhui Tian http://orcid.org/0000-0003-1577-9179
Qian Ba http://orcid.org/0000-0002-7253-7566
Ziliang Wang http://orcid.org/0000-0002-2702-5887

## FUNDING

| Funder | Grant(s) | Author(s) |
| --- | --- | --- |
| Shanghai Municipal Health Commission | 2022YQ066 | Ziqi Chen |
| National Natural Science Foundation of China | 82304777 | Ziqi Chen |
| Science and Technology Innovation Plan Of Shanghai Science and Technology Commission | 23YF1442800 | Ziqi Chen |
| Shanghai Medical Innovation and Development Foundation | WL-HBQN-2021004K,WL-QNPY-2022004K | Ziqi Chen |

## AUTHOR CONTRIBUTIONS

Ziqi Chen, Conceptualization, Writing – original draft, Writing – review and editing | Zhipeng Zhang, Formal analysis | Bei Ning Nie, Formal analysis | Wei Huang, Validation | Ying Zhu, Validation | Long Zhang, Validation | Meng Xu, Validation | Mengfei Wang, Validation | Chenyue Yuan, Validation | Ningning Liu, Validation | Xinyi Wang, Validation | Jianhui Tian, Conceptualization, Supervision | Qian Ba, Conceptualization, Writing – review and editing | Ziliang Wang, Conceptualization, Supervision, Writing – review and editing

## DATA AVAILABILITY

All information is included in the paper or supporting files. All sequencing data generated in this study have been deposited in the NCBI Sequence Read Archive (SRA)

under the accession numbers PRJNA731589 and PRJNA763023. These data are publicly accessible and can be retrieved using the provided accession numbers.

## ETHICS APPROVAL

Experimental and animal procedures in the current study have been approved by the research committee at Shanghai Municipal Hospital of Traditional Chinese Medicine and were carried out in accordance with the approved guidelines.

## ADDITIONAL FILES

The following material is available online.

### Supplemental Material

**Supplemental figures (mSystems01188-24-s0001.pdf).** Figures S1 to S4.

### Open Peer Review

**PEER REVIEW HISTORY (review-history.pdf).** An accounting of the reviewer comments and feedback.

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
