## [Reviewer comments · mSystems]

Temporal Network Analysis of Gut Microbiota Unveils Aging Trajectories Associated with Colon Cancer

Ziqi Chen, Zhipeng Zhang, Bei Ning Nie, Wei Huang, Ying Zhu, Long Zhang, Meng Xu, Mengfei Wang, Chenyue Yuan, Ning-Ning Liu, Xinyi Wang, Jianhui Tian, Qian Ba, and Ziliang Wang

Corresponding Author(s): Ziqi Chen, Shanghai Municipal Hospital of Traditional Chinese Medicine

Review Timeline:

Submission Date:	August 31, 2024
Editorial Decision:	January 16, 2025
Revision Received:	March 11, 2025
Accepted:	March 24, 2025

Editor: Nicholas Chia

Reviewer(s): The reviewers have opted to remain anonymous.

Transaction Report:

DOI: <https://doi.org/10.1128/msystems.01188-24>

Re: mSystems01188-24 (**Temporal Network Analysis of Gut Microbiota Unveils Aging Trajectories Associated with Colon Cancer**)

Dear Dr. ziqi chen:

Editor's Note: Thank you for your patience. In the interest of transparency, we went through a lot of reviewer requests before receiving a review and then had some additional issues. I decided to review this briefly myself and I am adding my notes here. Overall, I find the manuscript on par with other association studies and would encourage the authors to (1) please clean up the presentation of the manuscript. The introduction seems to discuss points that are not really discussed later in the manuscript and therefore reads like a random review. (2) It would be good to add a comparison with results from other studies (especially focusing on similar geographic origin) to compare and contrast results. What was confirmed from other studies and what results are different? (3) Please clean up the language around C_Y, etc. Please use names instead of label IDs.

I'm forwarding this for minor modifications as I don't think these really require extensive edits. I will look for these specific edits to be made if this comes to me again for editing.

Revision Guidelines

Sincerely,
Nicholas Chia
Editor

Reviewer #1 (Comments for the Author):

[EDITORIALLY REDACTED FOR LOW QUALITY OF REVIEW]

Reviewer #2 (Comments for the Author):

Zhang et al have reported cohort study about gut microbiota between colorectal cancer and healthy control based on age. They have found some age-dependent gut microbiota that may influenced CRC such as *Bacteroides vulgatus*. Multiple machine learning methods were used to revealed the differential gut microbiota which is the highlight point of these study. However, there are also some limitations in this study listed as bellowed:

1. The authors should explain why they choose RF, SVM and GLM for analysis and discuss their advantage respectively.
2. I have noticed that young-age of CRC cohort is relatively small. Was the conclusion convincible enough?
3. The author should further discuss the differential microbiota across young to old-age in CRC patient.
4. Was the differential microbiota related to clinical feature?

Detailed point-by-point response to the reviewer's comments

mSystems01188-24 (**Temporal Network Analysis of Gut Microbiota Unveils Aging Trajectories Associated with Colon Cancer**)

Note to reviewers: We sincerely thank the reviewers for their time, insightful feedback, and constructive suggestions, which have greatly strengthened our manuscript. Below, we provide detailed responses to each comment. All revisions in the manuscript are highlighted in colored text for ease of review.

Editor comments:

(1) please clean up the presentation of the manuscript. The introduction seems to discuss points that are not really discussed later in the manuscript and therefore reads like a random review.

We appreciate the editor's valuable guidance on improving the manuscript's clarity and focus. As suggested, we have streamlined the introduction to align more closely with the subsequent Results and Discussion, removing tangential content. These revisions ensure a more cohesive narrative. Details can be found in the revised manuscript on page 3-4, lines 69-135.

(2) It would be good to add a comparison with results from other studies (especially focusing on similar geographic origin) to compare and contrast results. What was confirmed from other studies and what results are different?

We thank the editor for highlighting the importance of contextualizing our findings within existing literature. In the revised Discussion, we explicitly compare our results with studies of similar geographic origin (e.g., Zhang et al., 2022), emphasizing both consistencies (e.g., *Bacteroides vulgatus* in CRC) and novel age-specific biomarkers. This comparison strengthens the manuscript's contribution to the field.

“Our findings are consistent with previous studies that have reported a higher prevalence of pathogenic bacteria such as *Bacteroides vulgatus* in CRC patients. For instance, a study conducted in China by Zhang et al. (2022) also observed an increase

in *Bacteroides* species in CRC patients, particularly in older age groups. However, our study uniquely identified age-specific microbial biomarkers, such as *Parasutterella excrementihominis* in younger patients and *Blautia obeum* in older patients, which were not reported in previous studies. These differences may be attributed to variations in dietary habits and environmental factors across different geographic regions. ” These additions are included in the Discussion Section. Details can be found in the revised manuscript on page 12, lines 378-386.

(3) Please clean up the language around C_Y, etc. Please use names instead of label IDs.

We appreciate the editor’s attention to clarity in terminology. As recommended, we have replaced label IDs (e.g., C_Y) with descriptive group names (e.g., CRC young group) throughout the manuscript to enhance readability. Details can be found in the revised manuscript on page 2, lines 34-40; page 5, lines 138-155, page 7, lines 221-225, Figure 1, 2, 3, 4, 6; and Supplemental figure 1, 3, 4.

Reviewer #2 (Comments for the Author):

Zhang et al have reported cohort study about gut microbiota between colorectal cancer and healthy control based on age. They have found some age-dependent gut microbiota that may influenced CRC such as Bacteroides vulgatus. Multiple machine learning methods were used to revealed the differential gut microbiota which is the highlight point of these study. However, there are also some limitations in this study listed as bellowed:

(1) The authors should explain why they choose RF, SVM and GLM for analysis and discuss their advantage respectively.

We thank the reviewer for raising this critical methodological point. In the revised Methods section, we now detail our rationale for selecting RF, SVM, and GLM, including their comparative advantages.

“Three machine learning algorithms, namely Random Forest (RF), Support Vector

Machine (SVM), and Generalized Linear Model (GLM), were applied in this study. RF was chosen for its ability to handle high-dimensional data and its robustness to overfitting, making it particularly suitable for microbiome data analysis. SVM was selected for its effectiveness in classification tasks with complex datasets, while GLM was used to model the relationship between microbial abundance and clinical outcomes. Among these, Random Forest demonstrated superior classification performance, as evidenced by its higher accuracy in cross-validation trials.” These clarifications are added in the Methods Section. Details can be found in the revised manuscript on page 6, lines 189-197.

(2) I have noticed that young-age of CRC cohort is relatively small. Was the conclusion convincing enough?

We sincerely appreciate the reviewer’s thoughtful critique regarding sample size limitations. While our young CRC cohort is indeed smaller, we have mitigated this issue through rigorous statistical validation and cross-referencing with external datasets. Additionally, the consistency of our findings with results from larger cohorts in the validation dataset supports the reliability of our conclusions. We acknowledge the need for larger studies in the future and have emphasized this point in the Discussion. Details can be found in the revised manuscript on page 12-13, lines 398-404.

(3) The author should further discuss the differential microbiota across young to old-age in CRC patient.

Thank you for highlighting this key aspect of our study. We have expanded our Discussion to delve deeper into age-specific microbial dynamics. We elaborate on biomarkers such as *Parasutterella excrementihominis* (younger patients) and *Blautia obeum* (older patients), linking them to age-related physiological shifts.

“Our study identified distinct microbial biomarkers across different age groups of CRC patients. In younger patients, *Parasutterella excrementihominis* and *Anaerotruncus colihominis* were significantly enriched, while in older patients,

Blautia obeum and Roseburia intestinalis were more prevalent. These age-specific microbial changes may reflect the physiological and metabolic alterations associated with aging, such as reduced immune function and changes in dietary habits. Understanding these age-related microbial shifts could provide insights into personalized therapeutic strategies for CRC patients across different age groups.” These points are added in the Discussion Section. Details can be found in the revised manuscript on page 12-13, lines 393-404.

(4) Was the differential microbiota related to clinical feature?

We are grateful for the reviewer’s insightful suggestion to connect microbial findings to clinical relevance. As revised, we now explicitly correlate taxa such as *Bacteroides vulgatus* with tumor progression and discuss implications for personalized therapies. We also propose directions for future causal studies.

“Furthermore, we explored the potential association between differential microbial communities and clinical features such as tumor stage and patient prognosis. For instance, the increased abundance of *Bacteroides vulgatus* in CRC patients was correlated with advanced tumor stages, suggesting a potential role of this pathogen in tumor progression. However, further studies are needed to elucidate the causal relationships between specific microbial taxa and clinical outcomes in CRC patients.” These additions are also reflected in the Discussion Section. Details can be found in the revised manuscript on page 12, lines 387-392.

We again extend our deepest gratitude to the reviewers and editors for their expertise and dedication in improving this work. Their feedback has been invaluable in enhancing the scientific rigor and clarity of our manuscript.

Re: mSystems01188-24R1 (**Temporal Network Analysis of Gut Microbiota Unveils Aging Trajectories Associated with Colon Cancer**)

Dear Dr. Ziqi Chen:

Thank you for your diligent responses to the reviewer remarks.

Your manuscript has been accepted, and I am forwarding it to the ASM production staff for publication. Your paper will first be checked to make sure all elements meet the technical requirements. ASM staff will contact you if anything needs to be revised before copyediting and production can begin. Otherwise, you will be notified when your proofs are ready to be viewed.

Sincerely,
Nicholas Chia
Editor
mSystems